# Combined DiI and Antibody Labeling Reveals Complex Dysgenesis of Hippocampal Dendritic Spines in a Mouse Model of Fragile X Syndrome

**DOI:** 10.3390/biomedicines10112692

**Published:** 2022-10-25

**Authors:** Luisa Speranza, Kardelen Dalım Filiz, Sarah Goebel, Carla Perrone-Capano, Salvatore Pulcrano, Floriana Volpicelli, Anna Francesconi

**Affiliations:** 1Dominick P. Purpura Department of Neuroscience, Albert Einstein College of Medicine, New York, NY 10461, USA; 2Department of Pharmacy, School of Medicine and Surgery, University of Naples Federico II, 80131 Naples, Italy; 3Institute of Genetics and Biophysics “A. Buzzati-Traverso”, C.N.R., 80131 Naples, Italy

**Keywords:** DiIC_18_, dendritic spines, excitatory synapses, synaptopodin, Fragile X Syndrome, *Fmr1* knockout mouse, hippocampus

## Abstract

Structural, functional, and molecular alterations in excitatory spines are a common hallmark of many neurodevelopmental disorders including intellectual disability and autism. Here, we describe an optimized methodology, based on combined use of DiI and immunofluorescence, for rapid and sensitive characterization of the structure and composition of spines in native brain tissue. We successfully demonstrate the applicability of this approach by examining the properties of hippocampal spines in juvenile *Fmr1* KO mice, a mouse model of Fragile X Syndrome. We find that mutant mice display pervasive dysgenesis of spines evidenced by an overabundance of both abnormally elongated thin spines and cup-shaped spines, in combination with reduced density of mushroom spines. We further find that mushroom spines expressing the actin-binding protein Synaptopodin—a marker for spine apparatus—are more prevalent in mutant mice. Previous work identified spines with Synaptopodin/spine apparatus as the locus of mGluR-LTD, which is abnormally elevated in *Fmr1* KO mice. Altogether, our data suggest this enhancement may be linked to the preponderance of this subset of spines in the mutant. Overall, these findings demonstrate the sensitivity and versatility of the optimized methodology by uncovering a novel facet of spine dysgenesis in *Fmr1* KO mice.

## 1. Introduction

Dendritic spines are small dendritic protrusions that represent the postsynaptic compartment of most excitatory synapses. These structures allow for the establishment of neural microcircuits, which are in turn refined by spine remodeling or stabilization. Spines display morphological diversity and a high degree of activity-dependent structural/functional plasticity. For example, large mushroom spines with a prominent head are associated with increased synaptic strength and the formation of more stable synapses. In contrast, thin elongated spines with a smaller head are relatively unstable and are readily modified in response to activity [1]. Alterations in spine density, morphology, and size underlie the changes in synaptic connectivity and strength associated with long-term potentiation (LTP) or depression (LTD), the cellular correlates of learning and memory [2]. Notably, variations in spine number and properties have long been linked to neurodevelopmental and neurodegenerative disorders highlighting the importance of studying spine characteristics [3,4].

Excitatory synapses are also highly heterogeneous at the molecular level, the functional implications of which are just beginning to emerge [5,6,7]. Different synapses can form onto different sub-regions of individual neurons and brain regions depending on the specific molecular components they contain [7,8]. These diverse synapses can also be differentially modified by activity which underlie learning and memory [9,10,11,12], behavioral states [13,14,15], and disease conditions [3,16,17,18,19,20,21]. The extent of this synaptic heterogeneity is just beginning to be appreciated and presents a unique challenge. Currently, little is known about the physiological remodeling and pathological alterations at individual subsets of synapses, defined by both structural features and molecular makeup [21,22,23]. Therefore, to begin addressing this question, we optimized a method combining conventional immunofluorescence with the use of the fluorescent dye DiIC_18_ (or DiI) for concurrent visualization of dendritic spine morphology and synaptic protein composition ex vivo. 

DiIC_18_ (1,1′-dioctadecyl-3,3,3′,3′-tetramethylindocarbocyanine perchlorate) is a lipid-soluble dye with weak fluorescence until incorporated in the membrane lipid bilayer, where it diffuses at a rate of 0.2–0.6 mm/day in fixed specimens and 6 mm/day in living tissue [24,25]. Once incorporated in the membrane, DiIC_18_ strongly and persistently labels the entire neuron including the dendritic spines [26,27,28]. Early attempts of combining DiIC_18_ staining with immunolabeling in brain tissue were hampered by the limited compatibility of DiIC_18_ with detergents commonly used for membrane permeabilization during immunolabeling (e.g., Triton X-100, saponin). Indeed, such detergents broadly extract membrane lipids inducing leakage and disappearance of DiIC_18_ from finer neurite structures [29]. Interestingly, recent evidence suggests that digitonin—a detergent-like compound that forms a complex with membrane cholesterol—better preserves DiIC_18_ staining while enabling the detection of abundantly expressed proteins including axonal neurofilaments and nuclear antigens [26]. However, there is still limited evidence validating the capacity of DiIC_18_ to be used in combination with immunolabeling to identify specific subsets of spines [30]. 

Here we optimized and applied a combined DiIC_18_-immunolabeling method to characterize the properties of spines in *Fmr1* knockout (KO) mice, a pre-clinical model of Fragile X Syndrome (FXS). FXS is the most common form of inherited intellectual disability, with a high incidence of autism, arising from transcriptional silencing of the X-linked *FMR1* gene [31,32]. We show that hippocampal spines in pyramidal neurons of juvenile *Fmr1* KO mice display an immature profile with an overabundance of thin and branched spines. The overabundance of immature protrusions is accompanied by an overall reduction in the density of mushroom spines, and smaller spine head width, compared to wild types (WT). Surprisingly, despite their overall decreased abundance, we find that a subset of mushroom spines that express the actin-binding protein Synaptopodin (Synpo) [33]—which marks the presence of a spine apparatus (SA) [34]—are over-represented in mutant mice. Altogether, these observations demonstrate the sensitivity of the optimized DiIC_18_-based method by revealing alterations in a specific subset of mature spines that contain a SA, a novel facet of spine dysgenesis in FXS.

## 2. Materials and Methods

### 2.1. Animals

All animal procedures were conducted following protocols approved by the Albert Einstein College of Medicine, in accordance with the Guide for the care and use of laboratory animals by the United States PHS (Ethic Committee Name: IACUC, approval code #00001117, approval date 22 November 2019). *Fmr1* KO and WT mice (FVB.129P2-*Pde6b^+^* strain; The Jackson Laboratories, Bar Harbor, ME, USA) were bred in-house. Mice were fed ad libitum and housed with a 12 h light/dark cycle. Experiments were carried out in juvenile mice (1 WT female, 2 WT males, 3 *Fmr1* KO males) at postnatal day (PND) 22. Experimental mice were generated by crossing heterozygous females with WT males and genotyped using the following oligonucleotides: oligo.1 GTGGTTAGCTAAAGTGAGGATGAT and oligo.2 GTGGGCTCTATGGCTTCTGAGG for KO, oligo.1 and oligo.3 CAGGTTTGTTGGGATTAACAGATC for WT genotype, respectively. 

### 2.2. DiIC_18_ Staining 

The mice were deeply anesthetized to effect by placing them in a chamber that was prefilled with isoflurane. After euthanasia by decapitation, brains were removed from the skull, and washed three times for 5 min in phosphate buffer (PB; 137 mM NaCl, 2.7 mM KCl, 10 mM Na_2_HPO_4_, 2 mM KH_2_PO_4_) pH 7.4 at room temperature (RT). Brains were fixed by submerging for 16 h at 4 °C in 4% paraformaldehyde (PFA) made in PB buffer as above. Fixed brains were washed three times with PB, ~5 min per wash. Coronal tissue sections 150 µm-thick were cut with a vibratome (Leica Biosystems VT1000S, Deer Park, IL, US). Brain sections were collected with a paintbrush and stored in PB in a multi-well plate kept on ice until use on the same day or stored for up to one week at 4 °C. Freshly sliced tissue sections were labeled with DiIC_18_ (crystals, cat. D3911; Thermo Fisher Scientific, Waltham, MA, US) as previously described [25,29]. Briefly, the tissue sections were transferred with a paintbrush onto a glass slide, covered with PB, and unfolded carefully with a paintbrush. After the removal of most of the PB, DiIC_18_ crystals were applied by gently touching the tissue surface with the tip of an 18-gauge needle covered with crystals. For this, the needle was inserted directly into the DiIC_18_ bottle to coat with the finest crystals whereas the largest crystals were removed by gently tapping the needle on the wall of the DiIC_18_ container. The dye was applied to regions of interest that included the hippocampus and parietal and prefrontal cortex. For the hippocampus, DiIC_18_ was applied with multiple consecutive touches throughout the region in order to dilute the crystals. This procedure prevents the deposition of excess amounts of crystals in one area and results in different labeling intensity throughout the region to allow for visualization of an individual hippocampal neuron more easily. Next, the tissue sections were very gently overlaid with PB (RT, ~300 μL) to prevent dehydration and incubated for 15 min at RT protected from light, allowing the crystals to settle onto the tissue. The labeled tissue was then transferred to a multi-well plate with additional PB and incubated at 4 °C, protected from light, for seven to ten days followed by image acquisition. 

### 2.3. Immunofluorescence

For combined immunofluorescence (2-step protocol), the tissue sections pre-stained with DiIC_18_ for the indicated time were incubated for 30 min at RT with 100 µg/mL digitonin (Sigma Aldrich, St. Louis, MO, US; cat. D141) dissolved in 3% bovine serum albumin (BSA) in PB. Next, the tissue was transferred to a glass slide and incubated for 12 h at 4 °C with primary antibodies diluted in 3% BSA in PB with 100 µg/mL digitonin and then washed three times for 5 min with PB. Secondary antibodies conjugated to Alexa Fluor 488 were diluted in 3% BSA/PB and applied for 3 h at RT. Tissue sections were then washed three times for 5 min with PB and mounted with a cover glass using ProLong (Cell Signaling Technology, Danvers, MA, US). 

### 2.4. Antibodies 

Antibodies used in this study include rabbit anti-Synpo (1:400; Synaptic Systems, Goettingen, DE; RRID:AB_887825), guinea pig anti-Synpo (1:400; Synaptic Systems, RRID:AB_10549419), guinea pig anti-VGluT2 (1:500; Millipore, Burlington, MA, US; RRID:AB_1587626), rabbit anti-Synaptoporin (1:200, Proteintech, Rosemont, IL, US; RRID:AB_2878022). Anti-rabbit and anti-guinea pig secondary antibodies conjugated to Alexa Fluor 488 (1:400) were obtained from Thermo Fisher Scientific.

### 2.5. Microscopy and Image Analysis 

Images were acquired with a Leica SP5 point-scan confocal microscope mounted with a 63× oil immersion objective (N.A. 1.4) using 3 or 4× zoom-in function. Images at 1024 × 1024 pixel resolution were acquired with scan speed set at 8 and pinhole configured to 1 Airy unit for each channel. Stacks of images were acquired with a 0.5 µm Z step and reconstructed with Fiji [35] using the maximum intensity projection method (MIP) in the Z-stack function. Analysis of dendritic spines and fluorescent signal overlap were conducted blind to genotype on merged 2D Z-stacks using Fiji. Dendrites and dendritic protrusions were outlined and measured with the segmented line tool. Spine density is expressed as the number of spines per dendritic length (in μm). Morphometric analysis of spine properties was carried out as previously described [30]. Briefly, the length (l), head (h) and neck (n) width of dendritic protrusions were manually traced. The length was measured from the edge of the dendritic shaft to the tip of the protrusions; head dimensions were measured at the point of maximum width. Dendritic protrusions were classified according to commonly accepted criteria as described previously [30], including mushroom spines (h » n, h/n > 1.5), thin spines (l > 1 µm, h/n < 1.5), stubby spines (l < 0.5 µm, h/n ≤ 1), filopodia (h = n; l > 3 µm, 0.1 < n < 0.4 µm) and branched cup-shaped spines (neck split into 2 sub-necks, each with a small head). To quantify Synpo-positive (*S*+) and Synpo -negative (*S*−) spines, a color-merged image of DiIC_18_ and Synpo signal was generated, and spines were counted with the Cell Counter plugin. Points of fluorescent signal overlap were identified with the Colocalization Threshold and Coloc2 plugins after background subtraction and quantified with the Cell counter tool.

### 2.6. Statistical Analysis

Data are reported as mean ± SEM; comparisons between two groups used unpaired *t*-test with Welch’s correction whereas comparison of multiple groups used ANOVA with Tukey’s posthoc test. Generation of graphs and statistical tests were carried out with Prism 8.1 (GraphPad, San Diego, CA, US). 

## 3. Results

### 3.1. DiIC_18_ Combined with Immunolabeling Enables Morphological and Molecular Characterization of Individual Spines 

In the past, DiIC_18_ has been successfully used for neuron tracing due to its ability to become incorporated into the plasma membrane and diffuse throughout the entire neuron [36,37,38,39]. In 2008, Matsubayashi and colleagues explored the possibility of combining DiIC_18_ staining with immunodetection of abundant cytoskeletal and nuclear antigens using digitonin to improve antibody penetration into brain tissue without substantial loss of DiIC_18_ signal [26]. Here, we report a simple and reliable optimized method to visualize individual spines and low-abundance synaptic proteins in brain tissue using DiIC_18_ staining in combination with immunofluorescence and high-resolution confocal microscopy. The method includes either a simple 1-step protocol limited to visualization of neurites or a 2-step protocol for combined detection of post- or pre-synaptic proteins (Figure 1). In the 1-step protocol, mouse brain tissue sections are incubated with DiIC_18_ for seven to ten days followed by the acquisition of image stacks by confocal microscopy (Figure 1a and Appendix A). Dendritic protrusions are well preserved and can be outlined and measured using Fiji, an open-source image-processing platform [35]. The density of dendritic protrusions of different morphology, alterations of which is a hallmark of neurodevelopmental and neurodegenerative disorders, can be determined by classification according to established morphologic criteria (for details see Section 2). In the 2-step protocol, brain tissue sections are first stained with DiIC_18_ for seven to ten days, then permeabilized with digitonin (100 µg/mL) and immunolabeled in a digitonin solution containing primary antibodies (Figure 1b) followed by incubation with appropriate fluorescent secondary antibodies in a 3% BSA/PB solution (for details see Section 2). The structural integrity of dendritic spines and membrane incorporation of DiIC_18_ are maintained after incubation with digitonin, as illustrated by the visualization of a subset of mushroom spines that express Synpo, a marker of the SA (Figure 1b).

The 2-step protocol can be applied to visualize presynaptic sites expressing different subsets of proteins, thus distinguishing individual synapses based on both morphological and molecular identity and allowing one to access the complexity and heterogeneity of synapses in the brain. Using this method we were successfully able to visualize a subset of dendritic spines with different microanatomy—presence vs. absence of a SA labeled with anti-Synpo—adjacent to presynaptic terminals expressing either Vesicular Glutamate Transporter 2 (VGluT2; Figure 2a,b; Appendix A) or Synaptoporin (Synpr; Figure 2a,b; Appendix A) enriched in hippocampal mossy fibers [40].

Altogether, these examples demonstrate the sensitivity and resolution afforded by combining DiIC_18_ staining with immunofluorescence for the characterization of excitatory synapses in native tissue.

### 3.2. Dysgenesis of Dendritic Spines in the Hippocampus of Juvenile Fmr1 KO Mice 

Abnormalities in the density and morphology of dendritic spines have been linked to dysfunctions in neuronal networks in neurodevelopmental disorders, including FXS [41,42,43]. Early studies in FXS patients indicated an overabundance of spines [44], however, analysis of *Fmr1* KO mice has produced conflicting results, in particular in the hippocampal region [45]. In the mature (>60 PND) hippocampus of *Fmr1* KO mice, either an increased density of dendritic protrusions [46,47], normal density [48], or subregion-specific differences were noted compared to WT [49].

Here, we used combined DiIC_18_ staining and immunofluorescence to examine the properties of excitatory spines in the hippocampus of *Fmr1* KO mice at PND22, when the main wave of synaptogenesis is concluded [50]. Spine properties were evaluated in hippocampal pyramidal neurons by quantitative analysis of high-resolution confocal images that enable visualization of fine morphological features (Figure 3a). Spine density was first calculated by considering the total number of protrusions per length of dendritic segment, regardless of morphology. Concordant with findings in the adult hippocampus [46], we found that the overall density of dendritic protrusions was higher in *Fmr1* KO mice compared to WT (Figure 3b; protrusions/μm mean ± SEM, WT 1.22 ± 0.031 n = 82 dendrites vs. *Fmr1* KO 1.34 ± 0.029 n = 59 from N = 3 mice per group; *p* = 0.006). Next, we evaluated dendritic spine maturation determined based on morphological features including length, head width, and presence of a discernible neck region (see Section 2 for details). The density of thin spines (spines/μm mean ± SEM, WT 0.82 ± 0.14 n = 82 vs. *Fmr1* KO 0.97 ± 0.035 n = 59), stubby spines (mean ± SEM, WT 0.18 ± 0.007 n = 82 vs. *Fmr1* KO 0.24 ± 0.008 n = 59) and filopodia (mean ± SEM, WT 0.03 ± 0.003 n = 25 vs. *Fmr1* KO 0.05 ± 0.007 n = 26) was uniformly higher in the *Fmr1* KO compared to WT (Figure 3c). In contrast, mushroom spines were significantly less abundant in the mutant (Figure 3c; mean ± SEM, WT 0.21 ± 0.009 n = 82 vs. *Fmr1* KO 0.14 ± 0.007 n = 59), representing ~10% of all dendritic protrusions compared to ~17% in WT (Figure 3d).

Our analysis includes aggregate data from CA1 and CA3 regions, but regional differences were noted in the hippocampus of adult *Fmr1* KO mice. Individual analyses of CA1 vs. CA3 revealed subtle differences in spine composition. In both regions, thin spines were similarly increased in *Fmr1* KO mice compared to wild type (CA1 mean ± SEM, WT 0.86 ± 0.059 n = 18 vs. *Fmr1* KO 1.047 ± 0.18 n = 20; CA3 WT 0.73 ± 0.068 n = 15 vs. *Fmr1* KO 0.97 ± 0.24 n = 24). Mushroom spines appeared less abundant in CA1 but not CA3 (CA1 mean ± SEM, WT 0.19 ± 0.016 n = 18 vs. *Fmr1* KO 0.12 ± 0.07 n = 20; CA3 mean ± SEM, WT 0.17 ± 0.02 n = 15 vs. *Fmr1* KO 0.15 ± 0.018 n = 24) although the difference did not reach statistical significance in the small sampled population (Appendix A). Similarly, the density of stubby spines appeared modestly, but not significantly, increased in *Fmr1* KO CA3 region, but not CA1, (CA1 mean ± SEM, WT 0.20 ± 0.019 n = 18 vs. *Fmr1* KO 0.21 ± 0.015 n = 20; CA3 mean ± SEM, WT 0.17 ± 0.023 n = 15 vs. *Fmr1* KO 0.23 ± 0.054 n = 24) (Appendix A). These subtle alterations in dendritic spines in *Fmr1* KO mice are concordant with previous findings [51] and overall congruent with the presence of regional differences in spine maturation in the mutant.

Previous analysis of cortical and hippocampal regions in adult *Fmr1* KO mice using Golgi staining identified a prevalence of elongated and tortuous spines deemed ‘immature’ in appearance [48,52]. To assess morphological spine properties in the hippocampus of juvenile mutant mice, we measured the length and head width of individual protrusions in high-resolution images of DiIC_18_-stained tissue (Figure 4a). In *Fmr1* KO mice, thin spines appeared significantly more elongated than in WT littermates (Figure 4b; WT 1.13 ± 0.023, n = 569 spines, *Fmr1* KO 1.21 ± 0.079 n = 480; *p* < 0.0008) whereas the head width of mushroom spines was comparatively smaller (Figure 4c; WT 0.64 ± 0.016 n = 690, *Fmr1* KO 0.60 ± 0.039 n = 276; *p* = 0.018). High-resolution imaging also enabled the visualization of an additional category of spines characterized by a cup-shaped head (Figure 5a), termed cup-shaped or branched spines [53]. Although relatively sparse, branched spines were observed more frequently in *Fmr1* KO mice compared to WT littermates (Figure 5b; spines/μm WT 0.003 ± 0.001, n = 69 dendrites, *Fmr1* KO 0.012 ± 0.003 n = 60; *p* = 0.0043) and were detected in ~28% of the dendritic branches examined compared to ~10% in WT (Figure 5c).

Thus, at the completion of synaptogenesis and prior to potential compensatory changes in adulthood, the hippocampus of juvenile *Fmr1* KO mice displays an overall increased density of dendritic protrusions. This abnormality is further compounded by pervasive spine dysgenesis, exemplified by the prevalence of elongated thin spines and branched spines and concomitant with the depletion of mushroom spines.

### 3.3. Surplus of Mushroom Spines Expressing Synpo in the Hippocampus of Fmr1 KO Mice

Mature mushroom spines of telencephalic regions are characterized by heterogeneous microanatomy, with ~20% harboring a SA. The SA is composed of folded smooth ER tubules intercalated by actin filaments and Synpo, an actin-binding protein that is required for the formation and maintenance of the SA [34]. The formation of a SA in the postnatal brain follows the developmentally regulated expression of Synpo protein, first detected at ~PND5 and reaching adult levels at ~PND20 [54,55]. Mushroom spines with Synpo/SA have higher synaptic strength [56] and longer lifetime than those in which Synpo/SA is absent [33].

Although only partly understood, a function of the SA is local regulation of calcium [57]. Moreover, the presence of Synpo/SA in spines was recently shown to be required for induction of group I mGluR-dependent long-term depression (mGluR-LTD) at hippocampal synapses [30]. Since abnormally enhanced mGluR-LTD [58,59] is an established phenotype of *Fmr1* KO mice, we applied DiIC_18_ staining in combination with immunolabeling with anti-Synpo (2-step protocol; Figure 1b) to examine the abundance of spines expressing Synpo/SA in the *Fmr1* KO hippocampus at PND22, when Synpo expression has stabilized (Figure 6a). Notably, we found that the relative abundance of Synpo-positive (S+) mushroom spines compared to total mushroom spines (Figure 6b; WT 0.21 ± 0.055 n = 18, *Fmr1* KO 0.33 ± 0.12 n = 20) was higher in *Fmr1* KO mice compared to WT. Thus, although juvenile *Fmr1* KO mice display a prevalence of immature spines and overall reduced density of mushroom spines, stable mushroom spines containing Synpo/SA are over-represented compared to WT.

## 4. Discussion

Alteration of excitatory synapses on dendritic spine is a hallmark of neurodevelopmental disorders including FXS, autism, and schizophrenia as well as of neurodegenerative disorders [3,4,60]. Here, we describe a sensitive method to facilitate rapid, in-depth investigation of the properties of spines in brain tissue. The method, relying on the combined use of the fluorescent dye DiIC_18_ and optimized conditions for in situ detection of low abundance synaptic proteins, offers several advantages. First, it does not require the generation of transgenic lines expressing reporters such as Thy1-GFP or transduction of fluorescent proteins via injections of lentivirus or adenovirus encoding fluorescent proteins, a procedure that is time-consuming and needs careful titration to yield sparse labeling for imaging. Second, the concurrent visualization of structural landmarks and protein expression/localization permits the identification of individual subsets of spines (e.g., spines containing the SA) which can be distinguished from one another and allow one to begin to dissect the vast heterogeneity of the intact brain. In addition, the method enables precise morphometric analysis of finer dendritic protrusions with complex morphology (e.g., branched spines). These critical features together with the rapid workflow provide significant advantages compared to time-consuming approaches such as Golgi staining that are not compatible with immunolabeling. Third, the approach is versatile and can be applied to any brain region and any animal model. Finally, and importantly, it can be used on fixed tissue sections that can be stored prior to labeling, thus providing an extended time window for analysis, crucial for translating experiments and results to human tissue. In this paper, we successfully demonstrate the sensitivity and applicability of this approach by completing an in-depth characterization of spines in the hippocampus of *Fmr1* KO mice, an animal model of FXS [61]. FXS is caused by a CGG expansion in the 5′-UTR of the *FMR1* gene, resulting in transcriptional silencing and downregulation/loss of the encoded FMRP protein [62]. FXS is part of a group of brain disorders termed ‘synaptopathies’ thought to arise from dysfunctions of synapse development and plasticity [42,63,64,65,66,67]. An overabundance of spines with immature morphology was observed in cortical and hippocampal regions of FXS patients [42,68] suggesting defects in excitatory synapse formation/maintenance. The *Fmr1* KO mouse recapitulates many manifestations of FXS and has been instrumental in understanding its molecular and cellular underpinnings. However, studies of spine dysgenesis in mutant mice have remained inconclusive [42,44,45,69] with reports of either overabundance of spines with immature morphology or lack of detectable alterations [45]. Such discrepancy, particularly notable in the hippocampus, was attributed to differences in methodology (Golgi staining vs. in vivo live imaging), brain area, and/or age under consideration.

In this study, we used the optimized labeling method to re-examine the properties of excitatory spines in the *Fmr1* KO hippocampus at PND22 (juvenile). This age was chosen because it is past the hippocampal critical period of plasticity, the main wave of synaptogenesis is concluded [48], and hippocampus-dependent memories can form [70]. We detected an overabundance of thin spines that also appeared abnormally elongated in the dendrites of pyramidal neurons of mutant mice. These alterations, suggestive of an *immature* state, are concordant with reports in FXS patients and several reports in cortical regions of the *Fmr1* KO mouse. Moreover, we found that mutant mice display an excessive number of branched cup-shaped spines compared to WT, an abnormality not previously noted. In the rat hippocampus, cup-shaped spines are rare at PND15 but more frequent in the adult [71], and their density increases in response to stimulation such as environmental enrichment and LTP [72,73,74]. Branched spines were mostly described in studies using reconstructions from electron microscopy images and are seldom considered in morphological analyses by Golgi staining or in vivo 2-photon imaging, likely due to limited resolution. With the advent of super-resolution microscopy, branched spines were shown to be dynamic [75] and endowed with complex organization of the actin cytoskeleton [76]. Individual branches of cup-shaped spines were reported to receive inputs from separate boutons [72,77]. Interestingly, multiple innervation of spines was recently shown to occur in the somatosensory cortex of PND10-15 *Fmr1* KO mice and linked to circuit hyperexcitability [78]. Although untested, it is possible that an increased abundance of branched spines in the juvenile hippocampus may be linked to the hyperexcitable network in mutant mice [79,80]. 

Concomitant to an overabundance of thin and branched spines, we detected a decreased abundance of mushroom spines that also displayed smaller heads than WT, a finding in agreement with reports by others [46,47,51]. Mushroom spines form strong synapses, as indicated by the correlation of head dimensions with higher synaptic strength and AMPA receptor content [81]. Such spines are stable and were shown in some cases to endure for weeks to potentially organismal lifetime. In particular, mushroom spines containing the SA were shown to be more stable, with longer lifetime [33] and higher synaptic strength [56] than those without. Unexpectedly, we found that mushroom spines with SA are more represented in hippocampal pyramidal neurons of *Fmr1* KO mice than in WT. This observation is concordant with findings in organotypic slice cultures in which thorny excrescences, the postsynaptic locus of dentate gyrus mossy fiber boutons-CA3 synapses, show augmented formation of the SA in *Fmr1* KO mice as determined by Synpo labeling [80]. Mushroom spines containing the SA are required for induction of mGluR-LTD at CA3-CA1 synapses, a form of plasticity that is abnormally enhanced in *Fmr1* KO mice [30,58,82,83]. Moreover, mGluR-LTD was found to induce loss of mushroom spines that do not contain a SA while sparing those in which the SA is present [30]. The enhanced density of mushroom spines with SA in *Fmr1* KO mice would be in line with mGluR-LTD enhancement and overall reduced abundance of mushroom spines in the mutant. Future experiments will determine whether overabundance of SA-containing spines is causally related to, and potentially precedes, abnormal mGluR-LTD in the mutant.

## 5. Conclusions

We report the development of an optimized methodology to investigate the morphological and molecular properties of excitatory spine synapses in brain tissue. The feasibility and advantages of the methodology are supported by the demonstration of its capability to identify abnormalities in the morphology and composition of spines in a mouse model of Fragile X syndrome. Amongst the observed defects, we detail the previously undetected prevalence of spines containing a spine apparatus that may be linked to aberrant synaptic plasticity in mutant mice.

## Figures and Tables

**Figure 1 biomedicines-10-02692-f001:**
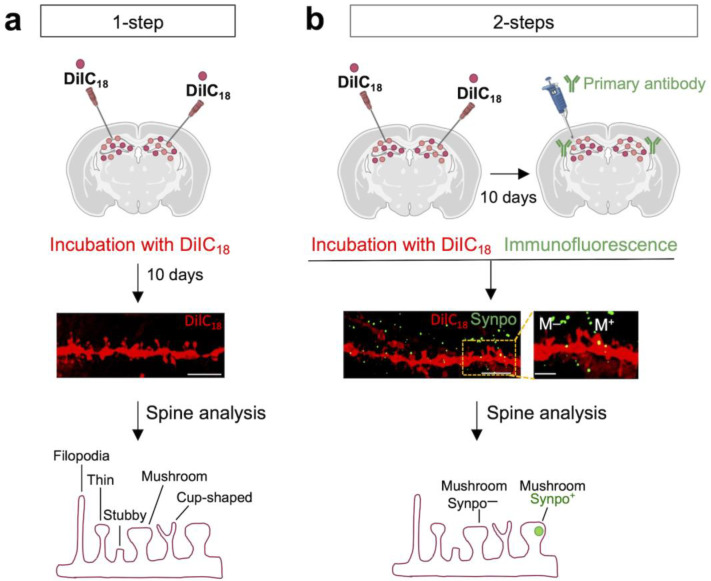
Workflow of DiIC_18_ staining combined with fluorescent immunolabeling. (**a**) Overview of the 1-step protocol for visualization and morphometric analysis of dendritic spines in rodent brain tissue. Shown in the middle panel is a representative confocal image of an area of the hippocampus from a coronal section of WT mouse brain stained with DiIC_18_; scale bar, 5 µm. The bottom panel is a graphical representation of different types of dendritic spines. (**b**) Overview of a 2-step protocol for morphometric analysis combined with detection of synaptic proteins. Shown in the middle panel is a representative confocal image of an area of the hippocampus from a coronal section stained with DiIC_18_ and immunolabeled with anti- Synpo antibody: scale bar 5 µm, magnified inset 2 µm. Mushroom (M) spines M−, Synpo-negative mushroom spine; M+ Synpo-positive mushroom spine. Illustrations in the top panels of (**a**,**b**) were created with Biorender.com (accessed on 25 August 2022).

**Figure 2 biomedicines-10-02692-f002:**
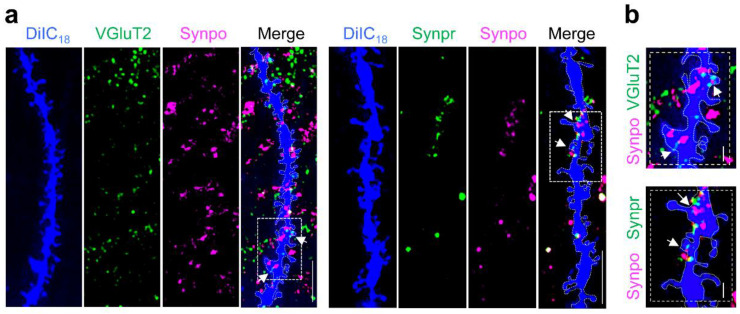
Visualization of Synpo-containing spines and VGluT2- or Synpr-positive terminals at hippocampal synapses ex vivo. (**a**) Representative confocal images of WT hippocampal dendrites stained with DiIC_18_ and labeled with anti-Synpo together with either anti-VGluT2 (left panels) or anti-Synpr (right panels). Shown are individual channels (DiIC_18_ in blue) and the overlay of the three channels (Merge); boxed areas are shown magnified in (**b**), scale bars 5 µm. (**b**) Magnified images of boxed areas in (**a**); scale bars, 2 µm. Arrows point to regions of Synpo/VGluT2 or Synpo/Synpr overlap.

**Figure 3 biomedicines-10-02692-f003:**
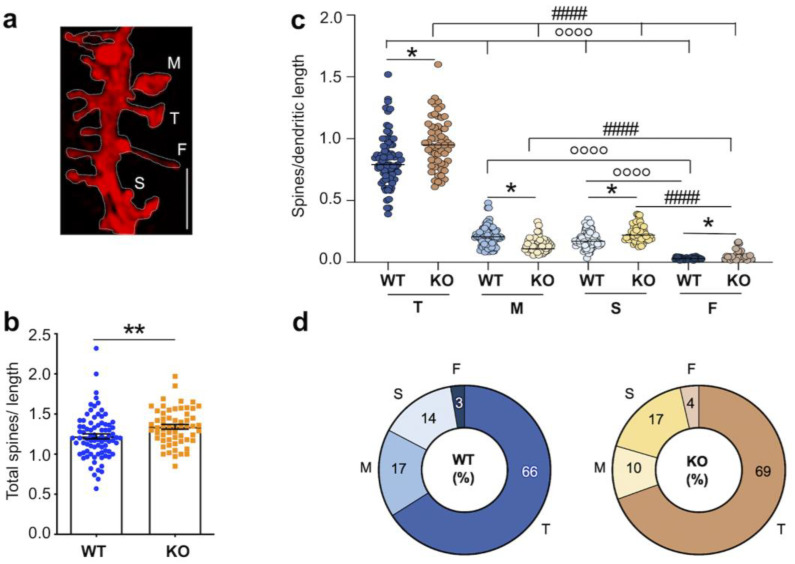
DiIC_18_ staining reveals dysgenesis of hippocampal spines in juvenile *Fmr1* KO mice. (**a**) Representative confocal image of a dendritic segment from WT mouse hippocampus, stained with DiIC_18_; scale bar, 2 µm. Labels indicate the different types of dendritic protrusions identified: M mushroom spines, T thin spines, S stubby spines, and F filopodia. (**b**) Quantification of total dendritic protrusions per dendritic length (µm) in WT and *Fmr1* KO mice. Differences were evaluated by Student’s *t*-test; ** *p* = 0.006, N = 3 mice per group. (**c**) Quantification of spines density per dendritic segment and categorization by morphology as thin, mushroom, stubby spines, and filopodia. Differences were evaluated by two-way ANOVA followed by Tukey’s post-hoc multiple comparisons test * *p* < 0.05 KO vs. WT, °°°° *p* < 0.0001 WT vs. WT, #### *p* < 0.0001 KO vs. KO. The effect of genotype (*p* = 0.0041) and spine’s type (*p* < 0.0001) is significant. The interaction between the main factors is significant (*p* < 0.0001); N = 3 mice per group. (**d**) Pie charts summarizing the relative proportion (%) of the four most common types of spines analyzed in (**c**).

**Figure 4 biomedicines-10-02692-f004:**
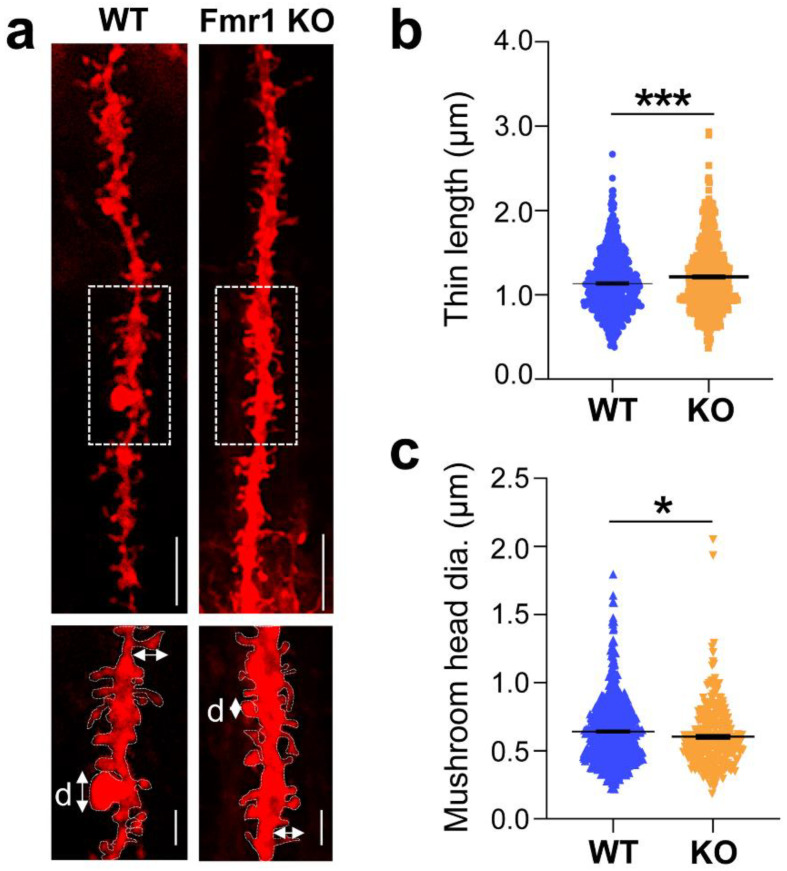
Morphological alterations of thin and mushroom spines in the *Fmr1* KO mouse hippocampus. (**a**) Representative confocal images of dendritic branches stained with DiIC_18_ from WT and *Fmr1* KO hippocampi; scale bars, 5 µm. Boxed regions are displayed in magnified insets below; scale bars, 2 µm. Arrows indicate the length and head width measurements; d, diameter of spine heads. (**b**) Quantification of the length of thin spines in WT and *Fmr1* KO littermates. (**c**) Quantification of mushroom spines head width (diameter) in WT and *Fmr1* KO mice. Differences were evaluated by Student’s *t*-test * *p* = 0.018, *** *p* < 0.0008, N = 3 mice per group.

**Figure 5 biomedicines-10-02692-f005:**
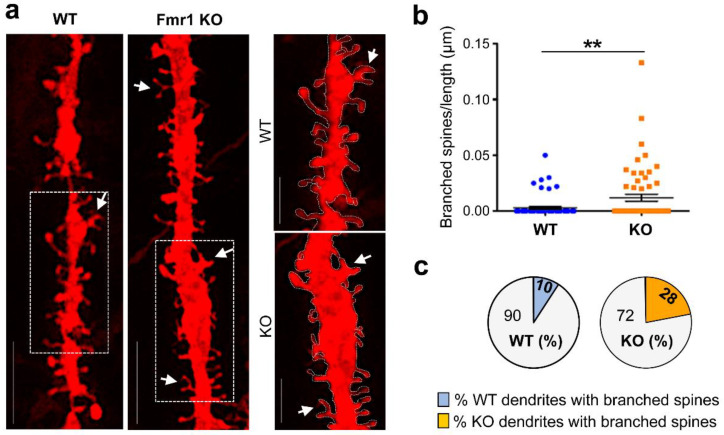
Abnormal prevalence of branched spines in the hippocampus of juvenile *Fmr1* KO mice. (**a**) Representative confocal images of dendritic branches stained with DiIC_18_ from WT and *Fmr1* KO hippocampi; scale bars, 5 µm. Boxed regions are displayed in magnified insets (right panels), scale bars, 2 µm. Arrows point to branched, cup-shaped spines. (**b**) Quantification of the relative density of branched (cup-shaped) spines per neurite length (µm) in WT and *Fmr1* KO mice. Differences were evaluated by Student’s *t*-test ** *p* = 0.0043, N = 3 mice per group. (**c**) Pie charts summarizing the relative proportion (%) of dendrites with branched spines relative to all dendritic branches examined in (**b**).

**Figure 6 biomedicines-10-02692-f006:**
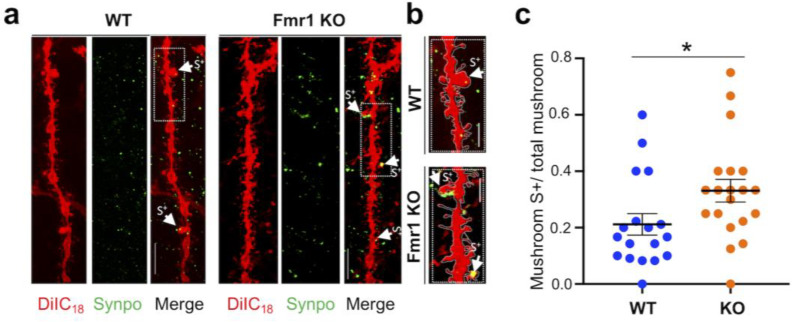
Increased abundance of mushroom spines containing Synpo in juvenile *Fmr1* KO mice. (**a**) Representative confocal images of dendritic branches stained with DiIC_18_ and immunolabeled with anti-Synpo from WT and *Fmr1* KO hippocampi; scale bars, 5 µm. (**b**) Magnified images of boxed areas in (**a**) i, scale bars 2 µm. Arrows point to Synpo-positive (S+) mushroom spines. (**c**) Quantification of Synpo-positive (S+) mushroom spines relative to the total number of mushroom spines in WT and *Fmr1* KO mice. Differences were evaluated by Student’s *t*-test * *p* = 0.038, from N = 3 mice per group.

## Data Availability

The data supporting the findings of this study are available upon request from the corresponding authors (A.F., L.S.).

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
