# Peer review of "Combined DiI and Antibody Labeling Reveals Complex Dysgenesis of Hippocampal Dendritic Spines in a Mouse Model of Fragile X Syndrome"

_biomedicines, 2022, doi:10.3390/biomedicines10112692_

Round 1

Reviewer 1 Report

Dear Authors,

I have read with interest your manuscript, describing a method for quick labeling of neuronal membranes to visualize dendritic spines and analyze their morphology via deposition of DiIC crystals on the surface of fixed tissue sections.

Before recommending the manuscript for publication, I require the following points to be addressed:

MAJOR points

- Line 64 "in vivo": please correct to "ex vivo". Using "in vivo" is misleading and suggests that experiments were performed on live mice.

- Lines 107-108: What were the composition and pH of PB? Was 4% PFA prepared in PB? Subtle changes in buffer composition can make a big difference in neuroanatomical methods.

- Figure 2: a line scan of signal intensity as a function of distance would help in formally appreciating the proximity (not overlap) between Synpo and vGluT2 or Synpr. According to the legend, right panels show Synpr, but are labeled as Synpo.

- Results §3.1: including some lower magnification images would be helpful in appreciating the degree of labeling sparsity and the average yield of the method.

- Figure 3c-d: What is the reason for splitting spine measurements between T-M and S-F? The densities of the different types of spines could be plotted in the same graph and tested as a whole using ANOVA-2, with main factors "genotype" and "spine type".

- Figure 3e: Possible differences in the frequency distribution of spine types should be tested using appropriate statistics.

- Lines 242-3: Were dendrites from CA1 and CA3 neurons pooled together in the analysis? What was the reason for this choice? The two populations should be analyzed separately (see also my comments on handling statistical "n"), as they received different presynaptic inputs, which can correspond to different postsynaptic parameters, including spine size and morphology. This is especially important also in light of the confusion about spine density in Fmr1 KO mice in the literature.

- Results §3.2-3.3, figures 3-4-5-6: As far as I can understand from the text, statistics were performed using individual neurons as the unit. I disagree with this approach; analyses should be repeated by using individual mice as the unit.

- Figure 5c: Possible differences in the proportion of dendrites with branched spines should be tested using appropriate statistics.

- Lines 355-76: Are the findings on spine density and shape consistent with published papers on the same brain area and animal age?

MINOR points

- Line 41, "Dendritic spines are the postsynaptic sites of excitatory synapses": please rephrase, as not all excitatory synapses are formed on dendritic spines.

- Line 134: please spell out the abbreviation "Synpo" at first appearance.

- Lines 49-50, "the cellular substrates of learning and memory": I strongly suggest changing "substrates" with "correlates", which more accurately reflects the experimental and conceptual values of activity-dependent synaptic plasticity data.

- Discussion: mentioning the pros of this method in comparison to alternatives (e.g., using transgenic lines like Thy1-GFP or transduction of fluorescent proteins via AAVs) would help the Reader in appreciating its potential.

- Line 358: "hippocampal-based" should be "hippocampus-dependent".

- Line 400: "capacity" should be "capability".

Author Response

POINT-BY-POINT RESPONSE
All text changes appear in red in the revised manuscript.

REVIEWER # 1

MAJOR points:

Concerns: Line 64 "in vivo": please correct to "ex vivo". Using "in vivo" is misleading and suggests that experiments were performed on live mice.

Response: We thank the reviewer for pointing this out. We replaced in vivo with ex vivo (lines 65 and 225).

Concerns: Lines 107-108: What were the composition and pH of PB? Was 4% PFA prepared in PB? Subtle changes in buffer composition can make a big difference in neuroanatomical methods.

Response: The composition and pH of the PB buffer is now stated in the Material and Methods section. We also specify that 4% PFA was prepared in PB buffer (lines 110-111).

Concerns: Figure 2: a line scan of signal intensity as a function of distance would help in formally appreciating the proximity (not overlap) between Synpo and vGluT2 or Synpr. According to the legend, right panels show Synpr, but are labeled as Synpo. 

Response: We thank the Reviewer for the suggestion. Appended below is a representative line scan for consideration (scale bar, 5μm). The scan illustrates signal intensity for DiIC18, VGluT2 and Synpo in function of distance. Legend in Figure 2 indicates Synpr and Synpo. 

Concerns: Results §3.1: including some lower magnification images would be helpful in appreciating the degree of labeling sparsity and the average yield of the method.

Response: We now provide a lower magnification image of neurons stained with DiIC18 in WT hippocampus (Figure S1).

Concerns: Figure 3c-d: What is the reason for splitting spine measurements between T-M and S-F? The densities of the different types of spines could be plotted in the same graph and tested as a whole using ANOVA-2, with main factors "genotype" and "spine type".

Response: We thank the Reviewer for the suggestion. We have amended the presentation of spine density and combined different categories in one graph (revised Figure 3c). We performed new statistical analysis testing genotype and spine type factor using two-way ANOVA as now indicated.

Concerns: Figure 3e: Possible differences in the frequency distribution of spine types should be tested using appropriate statistics.

Response: We appreciate the Reviewer’s comment and apologize for the lack of clarity. In Figure 3e (now Figure 3d) we summarized as percentages the data illustrated and quantified in Figure 3c. To clarify this point we amended the text in the figure legend (lines 252-253).

Concerns: Lines 242-3: Were dendrites from CA1 and CA3 neurons pooled together in the analysis? What was the reason for this choice? The two populations should be analyzed separately (see also my comments on handling statistical "n"), as they received different presynaptic inputs, which can correspond to different postsynaptic parameters, including spine size and morphology. This is especially important also in light of the confusion about spine density in Fmr1 KO mice in the literature.

Response: We thank the reviewer for the insightful comments. We pooled neurons from the CA1 and CA3 regions because the main objective was to demonstrate the sensitivity and flexibility of the Method using mutant mice as an example rather than in-depth analysis of the Fmr1 KO phenotype. As suggested, we have separated CA1 and CA3 regions and summarize the results in Figure S2. The analysis shows overall similar trends compared to the pooled data in CA1 whereas the difference in mushroom spines in CA3 is not significant. The method we present here can be used in the future for in-depth phenotypic characterization of spines and afferents labeled with different presynaptic markers.

Concerns: Results §3.2-3.3, figures 3-4-5-6: As far as I can understand from the text, statistics were performed using individual neurons as the unit. I disagree with this approach; analyses should be repeated by using individual mice as the unit.

Response: We appreciate the Reviewer’s comment and agree that analysis per individual mouse would be ideal. Based on observed mean difference and SD in mushroom spine density between genotypes, we calculated with power analysis that a minimum of 13 mice per genotype would be needed to obtain an alpha value of 0.05 at 95% CI. We produce littermates by crossing heterozygous females with wild type males and, depending on litter size, we do not always obtain wild type and mutant mice from the same litter. Therefore we estimate that we would need at least 13x2 litters to have sufficient animals with correct genotype/sex for the analysis. We did not embark on such large-scale study because the main objective of the manuscript was to demonstrate the validity of the method and the costs involved are fiscally beyond our means.  

Concerns: Figure 5c: Possible differences in the proportion of dendrites with branched spines should be tested using appropriate statistics.

Response: We appreciate the Reviewer’s comment and apologize for the lack of clarity. In Figure 5c we represented as percentage the data illustrated and quantified in Figure 5b. To clarify this point we amended the text in the figure legend (lines 317-319).

Concerns: Lines 355-76: Are the findings on spine density and shape consistent with published papers on the same brain area and animal age?

Response: We thank the Reviewer for raising this point. Our findings are congruent with work by others examining the hippocampus of P21 (FVB strain as in our work) and young adult (P60 in C57/Bl6 strain) mice. These studies also reported increased total and thin spines and decreased mushroom spines in the Fmr1 KO. Despite the overall agreement, values of spine density per micron and relative percentage of spine types vary across studies likely due to different methods of detection/visualization/categorization. See also response to Reviewer #2 (below). 

MINOR points:

Concerns: Line 41, "Dendritic spines are the postsynaptic sites of excitatory synapses": please rephrase, as not all excitatory synapses are formed on dendritic spines.

Response: We thank the reviewer for correctly pointing this out. We have now rephrased the sentence (lines 41-42).

Concerns: Point 2: Line 134, please spell out the abbreviation "Synpo" at first appearance.

Response: The abbreviation "Synpo" is now introduced in the abstract and at first appearance in the main text (line 31 and 90).

Concerns: Lines 49-50, "the cellular substrates of learning and memory": I strongly suggest changing "substrates" with "correlates", which more accurately reflects the experimental and conceptual values of activity-dependent synaptic plasticity data.

Response: We thank the Reviewer for the comment and amended the text accordingly (line 50).

Concerns: Discussion, mentioning the pros of this method in comparison to alternatives (e.g., using transgenic lines like Thy1-GFP or transduction of fluorescent proteins via AAVs) would help the Reader in appreciating its potential.

Response: We thank the Reviewer for the suggestion and now address this point in the Discussion (lines 361-365).

Concerns: Line 358, "hippocampal-based" should be "hippocampus-dependent".

Response: Text was amended as recommended (line 394).

Concerns: Line 400, "capacity" should be "capability".

Response: Text was corrected (line 436).

Reviewer 2 Report

The paper by Luisa Speranza and co-authors describes an implementation of a method based on the combination of DiI labelling of dendritic spines and immunofluorescence. They applied this method for the analysis of dendritic spines in the hippocampus of juvenile (PND22) WT and Fmr1 KO mice. They found that Fmr1 KO mice exhibit an overabundance of thin, stubby and branced spines and a lower density of mushroom spines compared with WT. In addition, they show that in Fmr1 KO mice there is an increased percentage of mushroom spines containing synaptopodin compared with WT mice.

The paper is well written and provide a useful protocol that can be applied to different model of neurological diseases. The finding that there are more mushroom spines containing synaptopodin is novel and adds a new piece of information regarding the pathophysiology of altered synaptic plasticity in Fmr1 KO mice.

I have a few concerns that need to be addressed.

1.     The authors always refer to spine synapses (Abstract, introduction lines 84) and not to spines. However, the authors analyze morphology of dendritic spines on DiI-labelled spines in WT and Fmr1 KO mice. I think that referring to spines synapses and not to spine only is misleading.

2.     Methods-euthanasia: how was it performed? Please describe.

3.     Methods -immunofluorescence: is the digitonin solution for the primary 12 hours incubation at the same concentration as in the blocking solution?

4.     Results- double labelling: it is difficult to judge the quality of the double-labeling staining. A good way to show the quality of the double-staining is performing a quantitation of the spines double stained with the anti-VGlut2 antibody. How many spines are double labelled with the presynaptic markers vGlut2? Similarly, how many of the Vglut2 spines are also double labelled with anti-synaptopodin?

5.     CA1 and CA3 regions receive a different set of afferents. Are there differences between the two regions in the density of spines and the morphological features of spines in WT and Fmr1 KO mice?

6.     The relative percentage of different types of spines reported here is different from what reported by others. In particular, the percentage of thin spines is much higher (66% WT, 69% KO) than that reported by others at similar ages (Pei et al., 2020; Jawaid et al., 2018). Furthermore, the percentage of mushroom spines reported here in WT and Fmr1 KO mice is much lower than previously reported (). More importantly, the reduction of number of mushroom spines and increase of thin and stubby spines in Fmr1 KO mice has been previously reported by others. Therefore, the relative papers must be cited and discussed (Pei et al., J Neuroscience 2020; Jawaid et al, Glia 2018).

7.     The number of mushroom spines containing synaptopodin is very low: between 0 and 0.01 in WT. Considering that it is estimated that about 20% of spines contain spine apparatus, as mentioned at page9 line 297, the number reported by the authors is very low and casts doubt about the reliability of the staining. See also point 4.

Author Response

POINT-BY-POINT RESPONSE

All text changes appear in red in the revised manuscript.

REVIEWER # 2

The paper is well written and provide a useful protocol that can be applied to different model of neurological diseases. The finding that there are more mushroom spines containing synaptopodin is novel and adds a new piece of information regarding the pathophysiology of altered synaptic plasticity in Fmr1 KO mice.

I have a few concerns that need to be addressed.

Concerns: The authors always refer to spine synapses (Abstract, introduction lines 84) and not to spines. However, the authors analyze morphology of dendritic spines on DiI-labelled spines in WT and Fmr1 KO mice. I think that referring to spines synapses and not to spine only is misleading.

Response: We thank the reviewer for raising this point. We have amended the title and main text to more closely align with the reported data.

Concerns: Methods-euthanasia: how was it performed? Please describe.

Response: We added the description in the Material and Methods section (lines 108-109).

Concerns: Methods -immunofluorescence: is the digitonin solution for the primary 12 hours incubation at the same concentration as in the blocking solution?

Response: We apologize for the omission. We added the information in the Material and Methods section (lines 137-140).

Concerns: Results- double labelling: it is difficult to judge the quality of the double-labeling staining. A good way to show the quality of the double-staining is performing a quantitation of the spines double stained with the anti-VGlut2 antibody. How many spines are double labelled with the presynaptic markers vGlut2? Similarly, how many of the Vglut2 spines are also double labelled with anti-synaptopodin?

Response: We thank the reviewer for the comment. We now provide a representative line scan for consideration (scale bar, 5μm). The scan illustrates signal intensity for DiIC18, VGluT2 and Synpo in function of distance. See image appended in page 1 of this document in response to Reviewer #1.

Concerns: CA1 and CA3 regions receive a different set of afferents. Are there differences between the two regions in the density of spines and the morphological features of spines in WT and Fmr1 KO mice?

Response: We thank the Reviewer for the suggestion. In response, we analyzed CA1 and CA3 regions separately and summarize the results in Figure S2. The analysis shows overall similar trends compared to the pooled data in CA1 (increased thin spines and reduced mushroom spines). In contrast, in the CA3 region thin spines are increased in the Fmr1 KO but mushroom spines not significantly altered.

Concerns: The relative percentage of different types of spines reported here is different from what reported by others. In particular, the percentage of thin spines is much higher (66% WT, 69% KO) than that reported by others at similar ages (Pei et al., 2020; Jawaid et al., 2018). Furthermore, the percentage of mushroom spines reported here in WT and Fmr1 KO mice is much lower than previously reported (). More importantly, the reduction of number of mushroom spines and increase of thin and stubby spines in Fmr1 KO mice has been previously reported by others. Therefore, the relative papers must be cited and discussed (Pei et al., J Neuroscience 2020; Jawaid et al, Glia 2018).

Response: We thank the Reviewer for bringing this issue to our attention. We cited Jawaid et al. 2018 in the original submission (ref#46) but were not familiar with Pei et al (2020) that we will now include (ref#47). Jawaid et al reported ~40 % thin spines in WT young adult mice (2 month-old; C57Bl6) visualized in CA1 by diolistic and 3D-EM, whereas Pei et al reported ~26 % (sub-region not specified) in WT P21 mice (FVB) visualized by Golgi staining. Differences in relative measurements are likely to depend not only on ages/background strain but also on sensitivity of labeling/detection method. Notwithstanding these differences, as the Reviewer points out, findings with our method are congruent with what reported in the above studies.

Concerns: The number of mushroom spines containing synaptopodin is very low: between 0 and 0.01 in WT. Considering that it is estimated that about 20% of spines contain spine apparatus, as mentioned at page 9 line 297, the number reported by the authors is very low and casts doubt about the reliability of the staining. See also point 4.

Response: We thank the Reviewer for bringing up this important point and apologize for the confusion. In the original Figure 6 the density of Mushroom Synpo+/ total mushroom was expressed per dendritic length (mm). In revised Figure 6b, to better compare with reports in the literature, we illustrate the fraction of mushroom spines that contain Synpo for both genotypes. As it can now be appreciated, in WT animals ~ 20% of mushroom spines contain Synpo/spine apparatus vs. ~ 30% in Fmr1 KO mice.

Reviewer 3 Report

Speranza et al improved a combined DiI and immunofluorescence staining, applied it on the characterization of spine synapses in a mouse model of Fragile X syndrome, and observed altered morphology of spines, in particular an increase in mushroom spines positive for synaptopodin, a marker for spine apparatus.

This study is well conducted, and the data are nicely presented.

To the reviewer, what is most significant and interesting in this article is the Method, rather than the Results. The authors are advised to describe the procedure of DiI staining in more detail, and to highlight the superiority of their method over the previous ones.

Author Response

POINT-BY-POINT RESPONSE

All text changes appear in red in the revised manuscript.

REVIEWER # 3

Concern: This study is well conducted, and the data are nicely presented.

To the reviewer, what is most significant and interesting in this article is the Method, rather than the Results. The authors are advised to describe the procedure of DiI staining in more detail, and to highlight the superiority of their method over the previous ones.

Response: We were very happy to read that the Reviewer found our work well conducted and the data nicely presented.  We appreciate the Reviewer’s comment that the most significant and interesting part is the Method described in the Article. We now provide a more detailed description of the method used for DiIC18 staining and immunofluorescence (page 3, 2.2 DiIC18 staining; 2.3 Immunofluorescence). We also highlight advantages of the method in the Discussion (lines 361-372).

Round 2

Reviewer 1 Report

Dear Authors,

I have read the revised version of your manuscript, in which most of my observations have been satisfactorily addressed.

Before the manuscript can proceed to the publication phase, I require the following points, corresponding to some revised parts of the manuscript, to be addressed:

- I suggest including "dendritic spines" in the title, not just "spines".

- Lines 110-1: Please check the composition of PB. The concentration values indicated in the text do not look correct.

- Figure 2: I could not find the linescan image in the main text nor in the supplementary material. Linescans should be enclosed for both Synpo/vGlut2 and Synpo/Synpr.

- Figure 3, lines 250-1: ANOVA-2, please indicate p values for each main factor or their interaction.

- Figure S2: please assess significant differences using ANOVA-2.

Author Response

Dear Authors,

I have read the revised version of your manuscript, in which most of my observations have been satisfactorily addressed.

Before the manuscript can proceed to the publication phase, I require the following points, corresponding to some revised parts of the manuscript, to be addressed:

POINT-BY-POINT RESPONSE

All text changes appear in red in the revised manuscript.

REVIEWER # 1

MINOR points:

Concerns: I suggest including "dendritic spines" in the title, not just "spines".

Response: We amended the title that now specifically states dendritic spines.

Concerns: Lines 110-1: Please check the composition of PB. The concentration values indicated in the text do not look correct.

Response: We thank the reviewer for pointing this out, and we corrected the composition of PB (lines 110-111).

Concerns: Figure 2: I could not find the linescan image in the main text nor in the supplementary material. Linescans should be enclosed for both Synpo/vGlut2 and Synpo/Synpr.

Response: We thank the Reviewer for the suggestion, the line scan analysis of vGluT2 originally presented in the rebuttal and a new line scan analysis of Synpr are now presented as supplemental Figure S2 and Figure S3 respectively.

Concerns: Figure 3, lines 250-1: ANOVA-2, please indicate p values for each main factor or their interaction.

Response: We thank the Reviewer for the suggestion, we added the p values for main factors and interaction in figure legend 3 (lines 251-253).

Concerns: Figure S2: please assess significant differences using ANOVA-2.

Response: Figure S4 (former Figure S2). As requested, we assessed differences using 2-way ANOVA now reported in the figure legend, and accordingly amended the text in the Results section (lines 287-296).

We found a statistically significant effect for spine type in CA1 and CA3. The effect of genotype alone in CA3 is significant whereas in CA1 did not quite reach significance (p=0.057), though interaction between the terms is significant in both cases. A Tukey post-hoc test revealed significant pairwise differences in thin spines between WT and KO in both areas. Pairwise individual difference for mushroom spines in CA1 and stubby spines in CA3 did not reach statistical significance (unlike with unpaired t-test analysis) possibly because of the unbalanced values across types in the multiple comparisons.

We would like to thank the reviewer for his/her constructive criticisms and insightful comments that helped us to improve our manuscript.